# Characterization of Novel SARM1 Inhibitors for the Treatment of Chemotherapy-Induced Peripheral Neuropathy

**DOI:** 10.3390/biomedicines12092123

**Published:** 2024-09-18

**Authors:** Jiayu Chen, Hao Li

**Affiliations:** State Key Laboratory of Bioreactor Engineering, Shanghai Key Laboratory of New Drug Design, and School of Pharmacy, East China University of Science and Technology, 130 Meilong Road, Shanghai 200237, China; cjybusiness@163.com

**Keywords:** SARM1, inhibitor, protective, CIPN

## Abstract

Background: Sterile α and Toll/IL-1 receptor motif-containing 1 (SARM1) is a central regulator of programmed axon death and a crucial nicotinamide adenine dinucleotide (NAD+) hydrolase (NADase) in mammalian tissues, hydrolyzing NAD+ and playing an important role in cellular NAD+ recycling. Abnormal SARM1 expression is linked to axon degeneration, which causes disability and disease progression in many neurodegenerative disorders of the peripheral and central nervous systems. Methods: In this study, we use PC6 assay of hydrolase activity, DRG axon regeneration and CIPN model to screen for potent SARM1 Inhibitors. Results: Two novel SARM1 inhibitors (compound 174 and 331P1) are charcterized for its high potency for SARM1 NADase. In a chemotherapy-induced peripheral neuropathy (CIPN) myopathy model, compound 331P1 treatment prevented the decline in neurofilament light chain (NfL) levels caused by axonal injury in a dose-dependent manner, associated with elevated intraepidermal nerve fiber (IENF) intensity in mouse foot paw tissue, suggesting its functionality in reversing axon degeneration. Conclusions: The newly designed SARM1 inhibitor 331P1 is a promising candidate due to its excellent in vivo efficacy, favorable CYP inhibition properties, and attractive safety profiles. The 331P1 compound possesses the potential to be developed as a novel neuroprotective therapy that can prevent or halt the neurodegenerative process in CIPN.

## 1. Background

Neurological diseases feature axon degeneration, resulting in peripheral neuropathy (PN) [1] traumatic brain injury (TBI) [2], and neurodegenerative diseases such as amyotrophic lateral sclerosis [3], Alzheimer’s disease [4], and Parkinson’s disease [5]. Axon degeneration, which separates the neurons from their targets, inducing the loss of neuronal function, is regarded as the main cause of neurological diseases [6]. Sterile α and Toll/IL-1 receptor motif-containing 1 (SARM1), with nicotinamide adenine dinucleotide (NAD^+^) hydrolase (NADase) activity [7], is the central executioner of the programmed axon destruction pathway [8]. Several mechanisms are involved in this scenario of calcium influx and, ultimately, axon fragmentation, which leads to specified local metabolic damage with ATP loss, mitochondrial depolarization defects, and motility dysfunction [9]. Moreover, genetic alteration of SARM1 prevents axon degeneration in axotomy in vivo [8,10], TBI [11], and glaucoma [12]. Notably, inhibition of SARM1 also protects axons from fragmentation in multiple peripheral neuropathies caused by chemotherapeutic and metabolic factors [13,14].

Chemotherapy-induced peripheral neuropathy (CIPN) is becoming a threat, with increasing impact when using potentially neurotoxic chemotherapy to deal with cancer and ensure survival. Several factors contribute to the increasing prevalence of clinical CIPN. The urgency and outcomes of CIPN are major unmet medical needs for which effective treatments for the prevention or reversal of CIPN are limited [15,16]. In recent years, there have been many reports of selective small inhibitors to address CIPN in preclinical studies, such as small molecular inhibitors of dual leucine zipper kinase [17,18] and SARM1 [19,20,21]. Although the detailed mechanisms differ, diverse chemotherapeutics could induce CIPN through a converged pathway, which is an active axon degeneration program, since SARM1 displays unique features in preventing CIPN and is functional in axon degeneration and potentially other neurological diseases. Herein, we discovered that a new compound, 331P1, specifically inhibits the NADase of SARM1 and plays pivotal roles in conferring neuroprotection in preclinical models of nerve injury and disease. Pharmacologic inhibition of SARM1 complexes results in improved neuroprotection, thus presenting these as therapeutic avenues for preclinical development.

## 2. Methods

### 2.1. Cell Lines

The open reading frame length 28–724 of hSARM1 expression that was constructed into HEK293T cells (hSARM1–HEK293T) was purchased from Cusabio Technology (Wuhan, China), and the sequence identity was verified using DNA sequencing. None of the cell strains used in this study were contaminated by mycoplasma. Dulbecco’s Modified Eagle Medium (Sigma-Aldrich, St. Louis, MO, USA; D0819) supplemented with 10% fetal calf serum, 2 μg/mL of puromycin, and a 1% penicillin–streptomycin solution was used for cell culturing, and the cells were maintained in a standard humidified tissue culture incubator with 5% CO_2_.

### 2.2. In Vitro Fluorescence Assays

A PC6 (synthesized according to reference [19]) in vitro assay was used to analyze the activity of SARM1 and CD38 (Cat# HY-P70731A). An enzymatic reaction mixture containing 50 mM PC6, 100 mM NAD^+^, and 100 mM nicotinamide mononucleotide (NMN) (only for SARM1 activation) in phosphate-buffered saline (PBS) was added after the enzymes were incubated. The fluorescence absorbance of the enzymatic reaction was tested in black 96-well plates (Corning, Somerville, MA, USA) using an Infinite M200 PRO microplate reader (Tecan, Kanagawa, Japan). For the screening assays, PC6 was incubated with the enzymes, and the kinetics of fluorescence production were measured at λex = 300 nm and λem = 410 nm. The initial rate of the reactions was quantified and analyzed using the slope of the fluorescence increase during the first few minutes.

### 2.3. Primary Neuronal Culture for the DRG Assay

Pregnant C57BL/6 mice were obtained from the Shanghai SLAC Center (Shanghai, China). On day 13.5 of embryonic development, dorsal root ganglia (DRGs) were dissected from wild-type (WT) mouse embryos and mechanically dissociated after incubation with 0.05% trypsin EDTA at 37 °C for 15 min. The DRG was applied with 1 mL of cell suspension onto a 24-well plate coated with poly-D-lysine/laminin and cultured in neurobasal medium (Gibco, Waltham, MA, USA; 21103049) supplemented with 2% B27, 50 mg/mL of nerve growth factor (2.5S, beta subunit), 0.45% D-glucose, 1% glutamine, 100 U/mL of penicillin, and 100 U/mL of streptomycin. Then, 5 mM 5-fluoro-20-deoxyuridine and 5 mM uridine were added to the culture medium to inhibit the growth of non-neuronal cells. Axonal resection was performed using a microscope knife at the stage of 9–12 days in vitro (DIV). Neurons were fixed in 4% paraformaldehyde at designated time points, washed with PBS containing 0.1% TritonX-100, blocked with wash buffer containing 1% bovine serum albumin, and stained with anti-tubulin β3 (Sigma, St. Louis, MO, USA; MAB5564) overnight. Axons were imaged on a Zeiss CD7 microscope (Oberkochen, Germany) using a plan apochromat 5×/0.35 objective, Optivar 23 tubulin, and a custom threshold method to identify intact and fragmented axons, thereby quantifying axonal degeneration. The programming was conducted using Zeiss Zen analysis software (ZEISS ZEN3.6).

A Leica DMi8 inverted microscope (Wetzlar, Germany) was used to capture in vivo phase contrast images of DRG neurons cultured in vitro at specified time points after axonal transection. The severity of neuronal degeneration was evaluated using the degeneration index, calculated as the ratio of the number of degenerated neuron pixels to the total number of neurons. Each image indicated the number of random, non-overlapping images of different DRG explants in each group from at least three independent replicates.

### 2.4. Imaging and Quantification of Axon Degeneration after Axotomy Treatment

During axonal transection, a DRG was seeded into a 24-well plate, and 5 mM 5-fluoro-2′-deoxyuridine and 5 mM uridine were added on another day. On DIV5, axons were pre-incubated with drugs for 0.5 h and cut at the proximal segment using a 3 mm flat blade under microscope guidance to remove cell bodies. For vincristine (VCR) treatment, DRG explants on DIV9-13 were cultured with 50 nM VCR and paclitaxel (PTX) with or without candidate drugs. Using an Olympus inverted optical microscope, approximately 9–12 axon images were captured in the bright field at each designated time point of processing using a 20× microscope. Axonal degeneration was quantified using ImageJ (https://imagej.net/, accessed on 6 August 2024). For each processing, 60 random grid squares of 147 pixels were cropped and binarized, and ImageJ’s particle analyzer module was used to quantify the total axonal area (size = 16 infinite pixels) and degenerate axons (size = 16–10000 pixels). The axonal degeneration index was calculated based on the ratio of degenerated axons to the total area of axons.

### 2.5. Sciatic Nerve Axotomy (SNA)

The mice received the indicated dose orally with compound 10 or a carrier (0.5% methylcellulose, 0.1% Tween 80). After 60 min, the mice were anesthetized with isoflurane/oxygen. When cutting the sciatic nerve, the skin was cut open in the middle of the thigh to expose the sciatic nerve and track it to the sciatic notch, taking care to minimize damage to surrounding tissues. The nerve was cut at a distance of 5 mm from the distal end of the incision, and a 1–2 mm portion was removed from the distal end to prevent reconnection at the end of the incision. Finally, fine surgical sutures were used to suture the skin. The animals were kept warm until they fully recovered. During the 15 h experiment, animals treated with the 174 and Nura compounds received a second dose 7 h after the first dose. A subcutaneous injection of 5 mg/kg of Karofen was administered to alleviate postoperative pain.

### 2.6. PTX CIPN Model

On days 1 and 6, the animals were placed under a heating lamp for 30 min. Then, the mice were intravenously injected with PTX (25 mg/kg) or a carrier (12.5% Cremophor VR/12.5% ethanol/75% physiological saline) through the tail vein using a venous catheter (29-gauge needle) at a dose of 10 mL/kg. Starting from 2 h before PTX administration, the animals were orally administered compound 331P1, Nura, or Disarm in a carrier (0.5% methylcellulose, 0.1% Tween 80) at doses of 200, 100, and 300 mg/kg, respectively, until the end of the experiment. The plasma was diluted at a ratio of 1:50 in the detection dilution buffer (Quanterix, Billerica, MA, USA) of the Simoa neurofilament light chain (NfL) detection kit, and then, NfL was measured using an HD-X analyzer (Quanterix) according to the manufacturer’s instructions.

Euthanasia was performed on the mice using carbon dioxide, followed by cervical dislocation. Mouse hind paw pads 3 and 4 were collected using surgical blades. The tissue was fixed in Zamboni fixative for 3–6 h and then gently shaken and frozen overnight in PBS containing 30% sucrose. The foot pads were embedded in 15% gelatin and frozen. Then, 20 µm sections were collected using a frozen slicer, loaded, cleaned, and stained with ubiquitin C-terminal hydrolase L1 or PGP 9.5 antibody (Sigma, St. Louis, MO, USA, AB5898) followed by fluorescently labeled secondary antibodies. Sections were imaged at 10× using MBF Bioscience’s modified Zeiss AxioImager microscope and ApoTome (Oberkochen, Germany).

### 2.7. Western Blots

Cells were lysed using a radioimmunoprecipitation assay buffer (50 mM Tris HCl, 150 mM NaCl, 1 mM EDTA, and 0.05% Triton, pH 7.4). Each sample was loaded onto 10–12% SDS-PAGE gel, and then the gel was transferred with protein onto a polyvinylidene fluoride membrane. The membrane was blocked with 5% milk and then imprinted with anti-SARM1 and anti-Tubulin (TransGen Biotech, Beijing, China) as an internal control. After incubation with horseradish-peroxidase-conjugated secondary antibodies, enhanced chemiluminescence (Abvansta, San Jose, CA, USA) staining was performed, and the Chemidoc MP system and ImageLab software (Bio-Rad, Hercules, CA, USA, https://www.bio-rad.com/zh-cn/product/chemidoc-mp-imaging-system?ID=NINJ8ZE8Z, accessed on 6 August 2024) were used for detection and quantification of proteins on the Western blot.

### 2.8. Cytochrome P450 Inhibition Assay

To evaluate the inhibitory potential of the test articles on CYP3A4 using human liver microsomes, a 96-well plate filled with cryopreserved human hepatocytes accommodated up to 9 test compounds (at three clinically relevant concentrations), as well as CYP1A2-, CYP2B6-, and CYP3A4/5-positive and -negative controls, and three different batches of cryopreserved human hepatocytes as vehicle controls (0.1% DMSO). Additionally, the positive controls included ritonavir for CYP3A inactivation/induction and staurosporine for cytotoxicity. Other compounds used as positive controls were 3-methylcholanthracene (a CYP1A2 inducer), phenobarbital (a CYP2B6 inducer), and rifampicin (a CYP3A4/5 inducer). The experimental procedure involved performing curve fitting to calculate the half-maximal inhibitory concentration (IC_50_) using a sigmoidal (non-linear) dose–response model (GraphPad Prism 5.0 or Xlfit model 205) based on the data calculated using the formula below.
*% of CYP inhibition = 100% − % of negative control

## 3. Results

### 3.1. High-Throughput Screening for SARM1 Inhibitors

Based on previous research [21,22] and their structure–activity relationships, several compounds were designed and synthesized in this study. To discover potent and effective hSARM1 inhibitors, a modified PC6 assay for a high-throughput screening method [20] was used (Figure 1A). The modifications introduced in the PC6 assay strengthened the capacity of stable fluorescent signals over time and resulted in high sensitivity for inhibition by the screened compounds, ensuring accuracy and diminishing false-positive readouts.

Several compounds were screened to verify their inhibition of recombinant hSARM1 (28–724) hydrolysis activity in the biochemical assay. The inhibition rate ranged from 17.2 to 56.8 nM of IC_50_ from the five selected compounds in the enzymatic screening (Figure 1C–G), except for the IC_50_ value 189.3 nM of the 331P1 compound (Figure 1B), which means that all of the newly designed compounds achieved robust inhibition of NAD^+^ hydrolysis activity. Moreover, the compound 331P1 demonstrated an equivalent inhibition capacity of SARM1 to the compound developed in [21] with an IC_50_ of 160 nM, possessing the strongest inhibition of its kind.

All six compounds (the structures are listed in Appendix A) were analyzed in a PC6 screening assay using human recombinant SARM1 28–724 that was pre-incubated with NMN for the activation of SARM1.

### 3.2. The Inhibition Preference of Compounds for SARM1 and Not for CD38

To assess the potent inhibition selectivity for SARM1 NADase, all of the isothiazoles were tested for inhibition against another NADase, CD38. The results show that none of the compounds had an inhibitory effect on the enzymatic activity of CD38 (Table 1), while the CD38 inhibitor (78c, Cat# HY-123999) displayed potent inhibition.

### 3.3. The Designed SARM1 Inhibitor Compounds Could Prevent Cell Death Caused by TIR (Toll/Interleukin-1 Receptor) Dimerization-Induced NAD^+^ Loss

To verify the inhibitory mechanism of SARM1 owing to the TIR domain, the SARM1 TIR domain was constructed into an Fkbp variant (DmrB) at its N terminus, which undergoes homodimerization in the presence of AP20187 (a rapamycin analog) (Figure 2A). TIR dimerization was enforced with AP20187; afterward, cell death was analyzed to monitor whether the compounds displayed anti-toxicity effects on the cells when incubated with AP20187.

The results demonstrate that the four compounds 331P1, 174, 109A, and 060 all preserved cell viability, which was lost in the model group containing AP20187 only. NR, an NAD^+^ precursor involved in the biosynthetic pathways that convert B3 vitamins into NAD^+^, was incubated with AP20187 in the culture, and the cell death caused by TIR dimerization was alleviated (Figure 2B).

### 3.4. The SARM1 Inhibitors Exhibited a Protective Function When Axons Were Exposed to Axotomy and VCR

To further verify the biochemical and structural observations of the substrate dependence of SARM1 inhibitors, co-treatment studies of inhibitors 174 and 331P1 were conducted in DRG axotomy- and VCR-induced DRG degeneration assays. Given our finding that the screened compounds displayed inhibition of NADase in the PC6 assay, we next sought to assess whether these pharmacological agents could prevent Wallerian degeneration. The compounds were tested in a cell-based assay using mouse DRG neurons and axons injured by axotomy, as previously described in the Methods Section.

In this study, embryonic day 13.5 (E13.5) mouse DRG explants were cultured in a neurobasal medium with mixing for at least seven days before inducing Wallerian degeneration. DRG axons were degenerated and almost eliminated 24 h post-axotomy (axotomized group) (Figure 3A and Appendix A). In contrast, the distal axons of the non-axotomized group remained smooth and intact. The results also suggested that 331P1 and 174 displayed better protection of axon degeneration than the other four compounds (Appendix A).

Afterwards, 0.2–5 µM SARM1 inhibitory compounds (331P1 and 174) were supplemented with the DRG medium right before axotomy. Both 331P1 and 174 at 5 μM showed strong protection against degeneration. 331P1 at 1 μM provided only mild protection, while at 0.2 μM, neither compound even slightly protected the axons (Figure 3A). We noted a significant improvement in the protection of DRGs from axotomy at increased concentrations of 174 and 331P1 (Figure 3B).

To study the protective effects of the SARM1 inhibitors on Wallerian degeneration after VCR administration, cultured DRG neurons were treated on DIV 10 with 50 nM VCR after long neurites were established and formed. Axon degeneration was quantified using immunofluorescence imaging and the axon degeneration index over an estimated time. After VCR administration, axons were completely fragmented within 48 h (Figure 3C).

In contrast, a 20 μM concentration of 331P1 or 174 displayed robust protection against axon degeneration when dosed prior to injury. Moreover, 331P1 and 174 at a 10 μM concentration conferred modest protection in the DRG degeneration assay (Figure 3C). We also quantified the degeneration index of axons administrated with the compounds (Figure 3D).

DRG neurons were treated with 50 nM of VCR, and axons were imaged using high-throughput automated imaging at indicated time points. The stain used was anti-tubulin β3. Quantification of fragmentation showed that axonal protection with 331P1 and 174 was dose-dependent (Figure 3B,D). Axon degeneration was quantified using a degeneration index ranging from 0 (perfectly intact) to 1 (perfectly fragmented). Values represent the mean ± SEM (n = 3/dose) and are representatives of three independent experiments with similar results.

### 3.5. The Expression Pattern of SARM1 Was Not Affected by Chemotherapy Drugs

The above research demonstrates that compounds 174 and 331P1 both displayed protective effects on axon degeneration after VCR and axotomy administration. In addition, the axon degeneration induced by PAC was also studied (Appendix A), and 331P1 showed potent protection as well as the Nura compound. To study whether SARM1 was activated and the potential role of SARM1 in degenerative diseases, the expression pattern of SARM1 in mouse DRG was first examined. The Western blot results show that the expression pattern of SARM1 was not upregulated or downregulated in the mouse DRG in either the PAC or VCR administration group (Figure 4). These results collectively suggest that SARM1 expression did not change in mouse DRGs when exposed to VCR or PAC.

### 3.6. SARM1 Inhibitors Prevented Nfl Release from Axotomized Nerves In Vivo

NfL is a biomarker for neurodegenerative diseases, and can be detected in both cerebrospinal fluid and blood. As an axonal cytoskeletal protein released into the circulation during traumatic axonal degeneration, NfL reflects degeneration and injury in axons, irrespective of the cause [23,24,25].

In our research, axonal degeneration after SNA could be monitored 15 h post-injury by measuring the plasma levels of NfL within injured nerves. To identify SARM1 inhibitors that could be rapidly progressed for testing in a neuropathy disease model, we evaluated selected compounds based on their ability to prevent increased NfL from being released into the blood induced by severed sciatic nerves at 15 h post-transection (Figure 5A). Additionally, a positive control, Nura, which has been verified to be protective, was introduced in the SNA model [21].

The mean baseline of the plasma NfL level was 95.7 ± 7.6 pg/mL (mean ± SEM) in the vehicle group from 12 independent cohorts. After 15 h of SNA, the mean plasma NfL level in the vehicle group increased to 3289.0 ± 154.2 pg/mL (mean ± SEM), i.e., a 34-fold increase compared to the baseline. Meanwhile, in the Nura and 174 compound treatment groups, the mean plasma NfL levels 15 h after SNA increased to 750.2 ± 9.2 pg/mL (mean ± SEM) and 662.3 ± 12.2 pg/mL (mean ± SEM), i.e., 8- and 7-fold increases from baseline, respectively. Thus, both the Nura and 174 compounds showed potent prevention of increased NfL at doses of 100 and 300 mpk, respectively, as well as improved functional outcomes for nerve injury in preclinical SNA models (Figure 5B).

### 3.7. SARM1 Inhibitors Protected Axons and Partially Prevented the Development of CIPN

Numerous studies have described how genetic loss-of-function of SARM1 can contribute to the protection of axons and prevent the occurrence of PN in preclinical CIPN models of VCR, PTX (paclitaxel, PAC in the above part), or bortezomib administration [13,26,27]. It has also been reported that, in the CIPN model, SARM1 KO mice have reduced mechanical allodynia compared to WT mice [20].

Herein, we introduced two positive controls, Nura and Disarm [20,21], verified to exhibit a protective function in VCR- and PTX-induced PN models, respectively.

To define the parameters of generating SARM1-dependent neuropathy, we evaluated 331P1 in a model of CIPN, in which C57BL6 mice were administered a maximum-tolerated dose of PTX of 25 mpk twice (Figure 6A). Along with using the biomarker NfL to evaluate the efficacy, the degeneration of intraepidermal nerve fibers (IENFs) was also adopted as a signature of PTX-induced damage to peripheral neurons.

In this disease model, after two doses of PTX, an early SARM1-dependent increase in plasma NfL release was observed (Figure 6B) from the mean baseline level of 204.7 pg/mL (mean ± SEM) in the vehicle group. In contrast, the mean plasma NfL level of the PTX group increased to 1032.8 pg/mL, i.e., a 5-fold increase, which suggests that the model was successfully constructed. Meanwhile, in the 331P1 treatment group, the mean plasma NfL level 15 days after PTX administration increased to 229.3 pg/mL, showing the robust inhibition of PTX-induced NfL release. The results show that 331P1 displayed a better protection effect than the control compounds of Nura and Disarm, which were 405.1 and 553.2 pg/mL, respectively.

Both the Nura and Disarm compounds showed slight protection against neuronal damage at the described doses, lowering plasma NfL and improving functional outcomes. Meanwhile, treatment with the 331P1 compound resulted in strong prevention of plasma NfL release (Figure 6B), displaying significant improvement in the protection of axons of distal nerve fibers after PTX administration (Figure 6D,E). Thus, inhibiting SARM1 during chemotherapy insult improved the outcomes in both the biomarkers of nerve injury and the nerve fibers associated with toxin-induced nerve damage.

Together, these results prove that administration with an orally bioavailable small-molecule inhibitor of SARM1-331P1 has the ability to retain axonal integrity and function, considering its effectiveness on the restoration of NfL loss and integrity of IENFs.

## 4. Discussion

SARM1 is the core executor of the axonal degeneration program. SARM1 is an NAD^+^ hydrolase that, when activated, reduces the NAD^+^ levels of axons, leading to metabolic crisis and axonal rupture. Knockout of SARM1 has a protective effect in CIPN [24,28] and TBI [11] models. Additionally, the genetic deletion of SARM1 protects against CIPN. Small-molecule inhibitors have been developed to inhibit the degeneration associated with CIPN, as they have been reported to protect injured axons [19,20,21,22].

In this study, a series of compounds for SARM1 inhibitors were designed based on previously reported compounds (Disarm and Nura), while compounds 174 and 331P1 were found to have an IC50 of 17.2 and 189.3 nM against human SARM1 hydrolase and possess excellent in vitro pharmacokinetic properties (Table 2, unpublished data).

These features make compounds 174 and 331P1 ideal candidates for further in vitro and in vivo studies. In a degeneration assay using DRG neurons, the results showed that both compounds possessed strong protection against axon degeneration when dosed prior to injury or VCR administration.

Some research groups have demonstrated the protective effect of small-molecule compounds in vitro on axons in DRG axonal transection models, while others have demonstrated the protective effect of high-dose early covalent inhibitors on axons, but their site of action is unknown [20,29]. In this study, we observed that these irreversible SARM1 inhibitors reduce the release of plasma NfL from damaged sciatic nerves during in vivo SNA. The results also showed that compound 331P1, as well as being a positive control, was sufficient to completely attenuate increases in plasma NfL levels and prevent the loss of intraepidermal nerve fibers induced by PTX in the CIPN model, thus contributing to the protection of the nerve axons.

This study fully evaluated compound 331P1 as a candidate for further development by assessing its potency, physiochemical properties, safety, and in vivo efficacy. We expect that the SARM1 inhibitor 331P1, with different scaffolds and modes of action, as well as better pharmacokinetic profiles, will be ideal for progressing to the clinical stage.

## 5. Summary and Limitations

A series of compounds for SARM1 inhibitors were designed based on the reported compounds, and compound 331P1 was identified with an IC50 of 189.3 nM against human SARM1 hydrolase and possessing in vitro pharmacokinetic properties. These features made compound 331P1 an ideal candidate for further in vivo studies. With compound 331P1, the protection of degenerated axons was verified in the DRG model. The efficacy of compound 331P1 was also tested in SNA and CIPN mouse models; the results showed that compound 331P1 restored plasma NFL levels, as expected for a SARM1 inhibitor. The mice with PAC-induced CIPN treated with compound 331P1 showed recovery of intraepidermal nerve fibers. These findings indicate that compound 331P1 is a selective and orally bioavailable SARM1 inhibitor with the potential for treating CIPN and other metabolic diseases associated with activated hydrolase activity of SARM1.

In conclusion, the most compelling evidence for the clinical translatability of compound 331P1 is the demonstration of safe and efficacious neuroprotection in in vivo models of nerve injury. In this study, the efficacy of compound 331P1 was prioritized for further evaluation of its in vivo protection.

However, more research is needed to better understand the detailed molecular mechanism of downstream effectors, as well as more direct evidence about whether the inhibitory effects of 331P1 are due to SARM1 inhibition. When it comes to 331P1 displaying potent protective effects against CIPN, it was found that calpain seems to be required for SARM1-dependent neuropathy in CIPN induced by both VCR and PAC [14,21,22,26]. Deciphering whether calpain contributes to the protective effect of 331P1 in CIPN awaits further study.

## Figures and Tables

**Figure 1 biomedicines-12-02123-f001:**
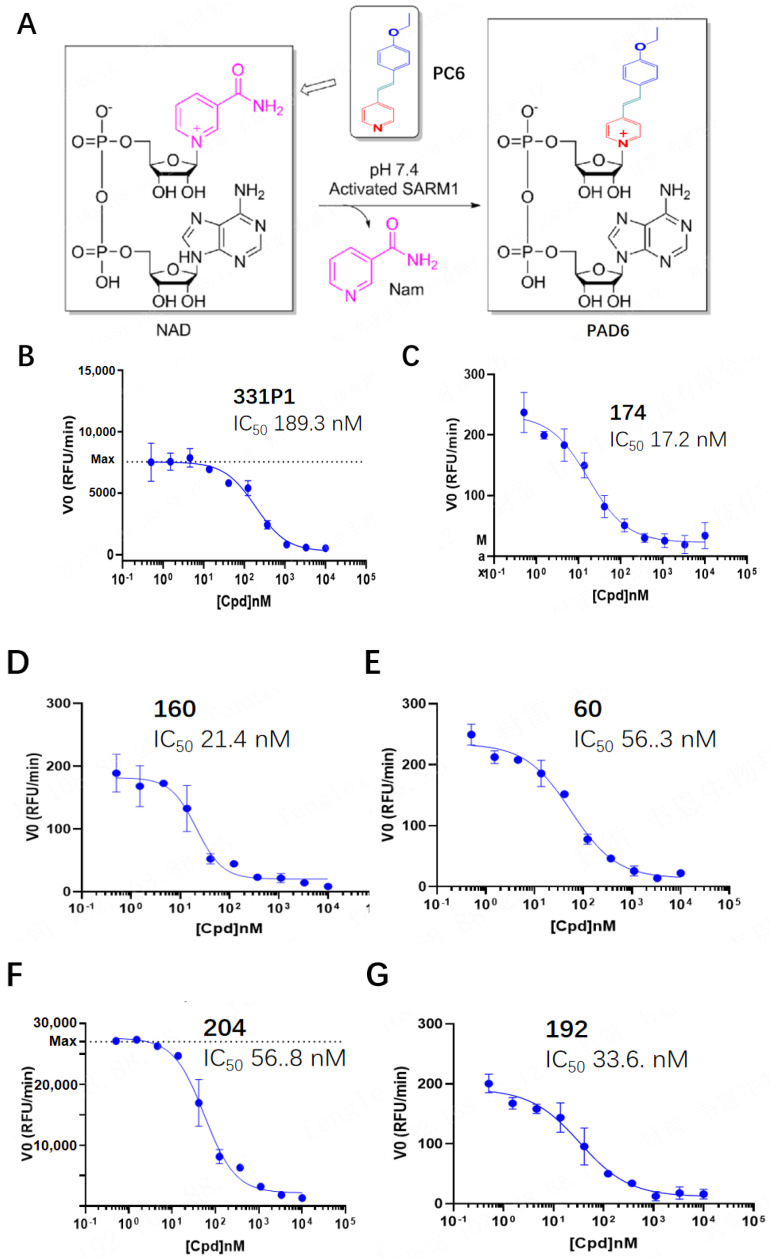
Design and characterization of PC6 probes for screening the potential of SARM1 inhibitor compounds. (**A**) The process of fluorescent imaging of the activated hSARM1 in a PC6-based screening assay. (**B**–**G**). Dose–response curves for compound screening of SARM1 inhibitors in a biochemical, high-performance PC6-based assay using hSARM1 28–724. Data points represent the mean ± standard deviation (SD) of three experiments.

**Figure 2 biomedicines-12-02123-f002:**
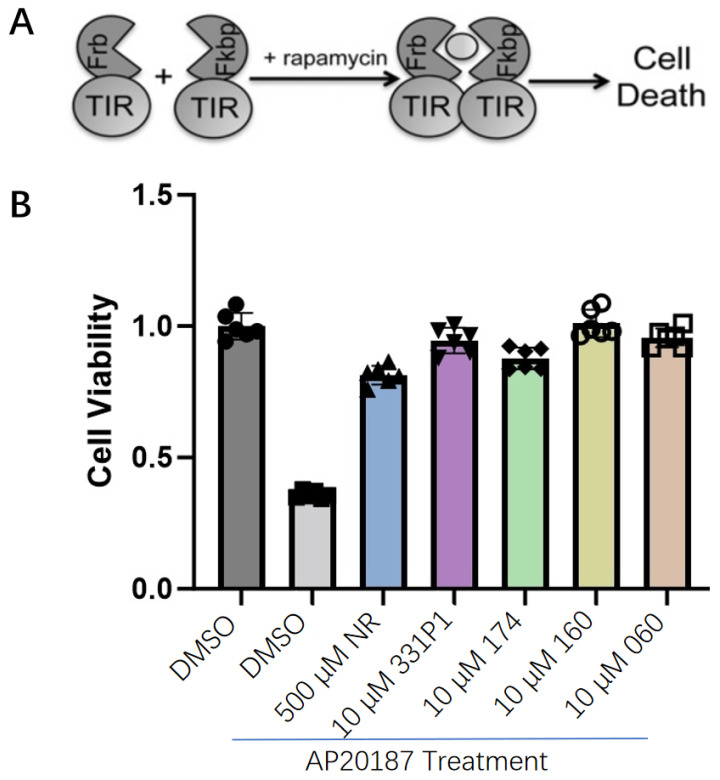
Design and characterization of the toxic effects of the SARM1 TIR domain in 293T cells using different compounds. (**A**) Scheme for TIR dimerization to verify the TIR domain using an Frb–Fkbp moiety fusion [22]. (**B**) Compounds used and NR display prevention of DmrB–TIR-dependent death 24 h after AP20187 addition.

**Figure 3 biomedicines-12-02123-f003:**
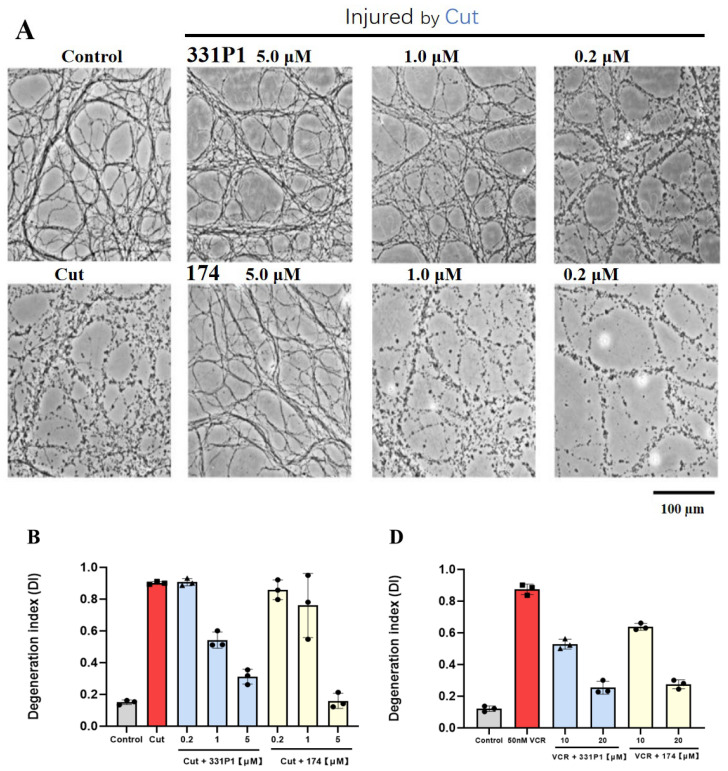
Design and characterization of the protective effects of injured axons in vitro. (**A**) Axons from DRG neurons were transected, and the degeneration of distal axons was monitored over the concentration of selective compounds 331P1 and 174. (**B**) Quantification of degeneration idex (DI) in fragmented axons induced by axotomy when ad-ministrated with 174 and 331P1. (**C**) Representative images of the control and SARM1-inhibitor-treated axons at the indicated concentration as VCR was added to the axon DRG. (**D**) Quantification of degeneration idex (DI) in fragmented axons induced by 50 nM VCR. Values represent mean ± SEM n = 3/dose. Representative of three independent experiments with similar results.

**Figure 4 biomedicines-12-02123-f004:**
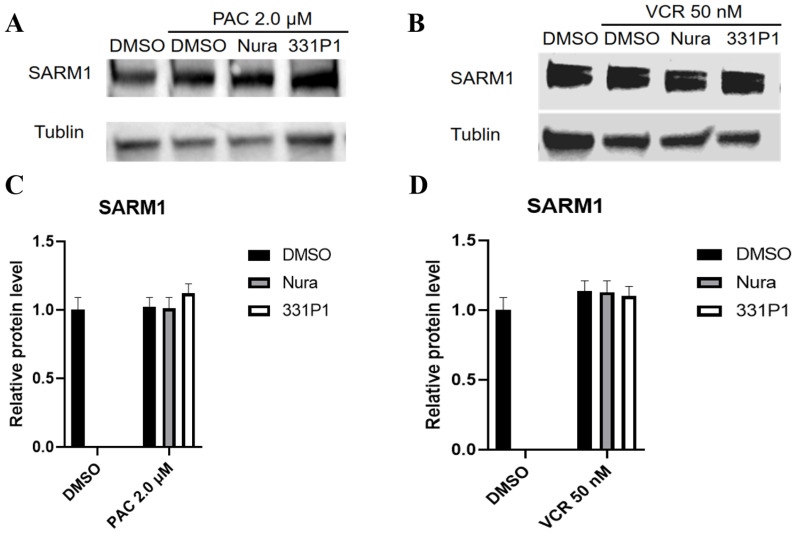
SARM1 expression did not change in DRGs when exposed to paclitaxel (PAC) (**A**) or vincristine (VCR) (**B**). When cells were exposed to PAC or VCR, the SARM1 protein was examined using Western blot analysis. (**C**,**D**) Quantification of of SARM1 expression in degenerated axons induced by PAC and VCR. Values represent mean ± SEM n = 3/dose.

**Figure 5 biomedicines-12-02123-f005:**
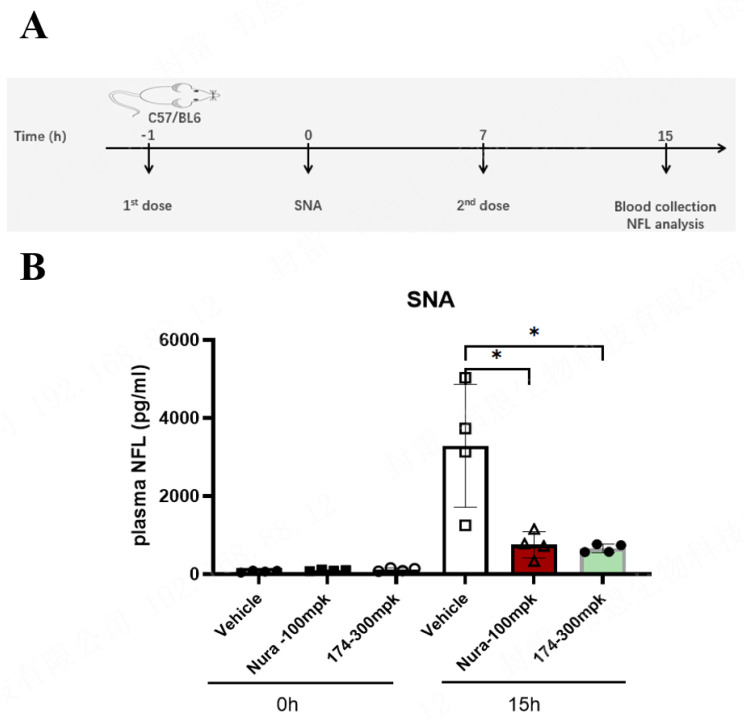
Design and characterization of the protection of SARM1 inhibitors in vivo in sciatic nerve axotomy (SNA). (**A**) SNA experimental design of mice with SARM1 inhibitor compound treatment. Mice were treated with the SARM1 inhibitors Nura compound and 174 at the doses indicated and subjected to SNA 1 h after the first dose. A second dose was administered 7 h after the first dose. NfL levels were measured in plasma. (**B**) Quantification of plasma NfL levels in the vehicle- or Nura compound- and 174-treated (n = 4) mice before (0 h) and after (15 h) SNA. Data represent the mean NfL threshold. Bars represent the mean ± SEM. Each symbol represents data from an individual mouse (n = 4). * *p* ≤ 0.05.

**Figure 6 biomedicines-12-02123-f006:**
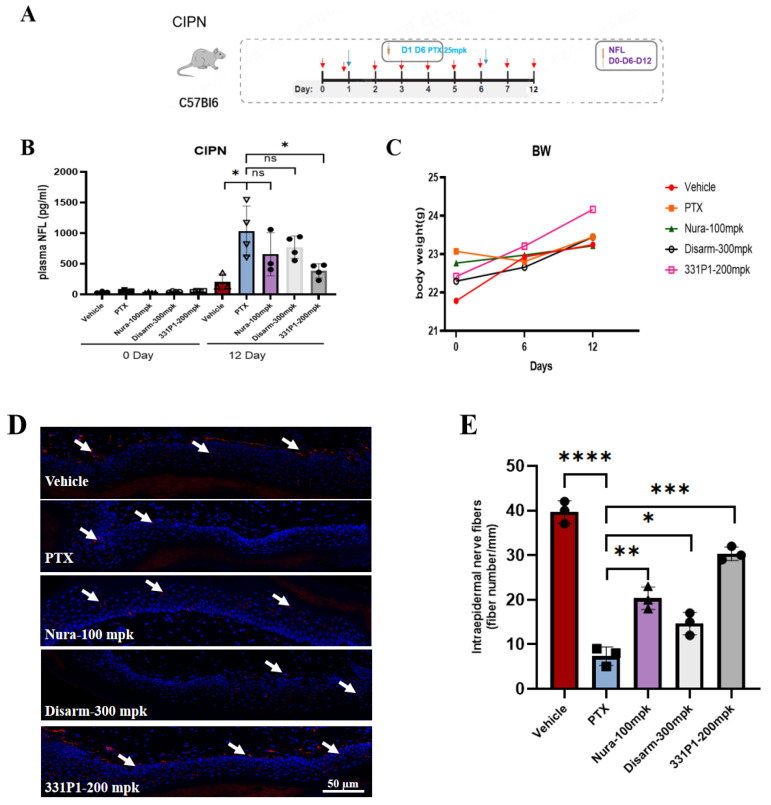
331P1 protects peripheral sensory neurons from PTX-induced degeneration. (**A**) Illustration of the experimental design of the CIPN model. Blue arrow denotes PTX administration at Day 1 and Day 6, red arrow denotes compound administration daily. (**B**) Quantification of plasma NfL levels in the vehicle- or Nura compound-, Disarm compound-, and 331P1-treated (n = 4) mice before (day 0) and after (day 12) PTX administration. Data represent the mean sensitivity threshold. Bars represent the mean ± SEM. Each symbol represents data from an individual mouse (n = 4). (**C**) Quantification of body weight in the vehicle- or Nura compound-, Disarm compound-, and 331P1-treated (n = 4) mice before (day 0), during (day 6), and after (day 12) PTX administration. (**D**,**E**) Representative images of the vehicle- or Nura compound-, Disarm compound-, and 331P1-treated (n = 4) mice’s footpad IENF and density quantification. Stain: anti-ubiquitin C-terminal hydrolase L1. White arrow denotes nerve fibers. Data represent the mean ± SD or SEM. * *p* ≤ 0.05, ** *p* ≤ 0.01, *** *p* ≤ 0.001, **** *p* ≤ 0.0001, ns denotes not significant.

**Table 1 biomedicines-12-02123-t001:** Enzymatic inhibition of CD38 by the compounds.

Compound	IC_50_ (μM)
331P1	>10
174	>10
60	>10
204	>10
160	>10
78c	0.004

**Table 2 biomedicines-12-02123-t002:** CYP induction of the selected compounds.

Compound (cpd)	CYP Inhibition	
	3A4 (Midazolam)	3A4 (Testosterone)
Disarm cpd	88.3%	90.0%
Nura cpd	74.1%	61.9%
331P1	−1.1%	−7.5%
174	NA	50.0%

## Data Availability

Data are contained within the article and Appendix A.

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
