# Peer review of "Characterization of Novel SARM1 Inhibitors for the Treatment of Chemotherapy-Induced Peripheral Neuropathy"

_biomedicines, 2024, doi:10.3390/biomedicines12092123_

Round 1

Reviewer 1 Report

Comments and Suggestions for Authors

The authors evaluated several compounds which may inhibit SARM1. Although the study shows that compound 174 and 331P1 reduce nerve damage or axon death, I have several concerns. First, I found that the methods are poor. The method should be described with sufficient detail to allow others to replicate and build on published results. Second, I’m not sure whether the protective effects of the compounds are through SARM1. The authors need to show that the effects are due to SARM1 inhibition. Third, there are many abbreviations without definition. It is very hard for me to understand the abbreviations. Lastly, the proposed presentation does not respect the elementary rules of a scientific writing.

Major concerns

#1. I found that the methods are poor. The method should be described with sufficient detail to allow others to replicate and build on published results.

#2. I’m not sure whether the protective effects of the compounds are through SARM1 (Figure 3, 5, and 6). The authors need to show that the effects are due to SARM1 inhibition.

#3. There are many abbreviations without definition (DLK, ORF, PC6, NfL, RIPA, PVDF, HRP, and SNA). It is very hard for me to understand the abbreviations. As the instruction for Authors says “Acronyms/Abbreviations/Initialisms should be defined the first time they appear in each of three sections: the abstract; the main text; the first figure or table.”

#4. Why did the author select 174 and 331P1 from 6 compounds (Figure 1)?

#5. Figure 4 does not prove anything. The authors need to show data from several doses of paclitaxel and vincristine. Also, please show quantitative data. It is very hard to compare expressions in Figure 4.

#6. I do not understand Table 2. I do not see the methods how the authors analyzed data (Methods section).

#7. Discussion is also poor. There is no summary and limitations. The proposed presentation does not respect the elementary rules of a scientific writing. Additionally, the manuscript would benefit tremendously from language editing by either a native English speaker or a professional editor.

Minor comments

#1. The title is misleading. The authors evaluated some compounds not “a” compound.

#2. There are two definitions of degeneration index (Figure 3). “Axon degeneration index was calculated as the ratio of the degeneration axon over total axon area” (Page 3, Line 100) “Axon degeneration was quantified using a degeneration index, which ranges from 0 (perfectly intact) to 1 (perfectly fragmented)” (Page 10, Line 230). Which is correct?

#3. What is compound “109A” (Page 6, Line 182)?

#4. Please show data low concentration to high concentrations (left to right) (Figure 3B).

#5. The authors need to show time course of NfL in Figure 5. Why did the authors select 15h?

#6. There is no method of table1.

Comments on the Quality of English Language

The manuscript would benefit tremendously from language editing by either a native English speaker or a professional editor.

Author Response

Major concerns

#1. I found that the methods are poor. The method should be described with sufficient detail to allow others to replicate and build on published results.

Response 1: Thank you for pointing this out. We agree with this comment. Therefore, l have rechecked and rewritten the whole methods in the revised manuscript.

#2. I’m not sure whether the protective effects of the compounds are through SARM1 (Figure 3, 5, and 6). The authors need to show that the effects are due to SARM1 inhibition.

Response 2:Sorry for the misunderstanding, as we have emphasized in revised manuscript (Page 4, Line 158). The structure of all the six compounds in this research are originated from the compounds in reference 21-22. The detailed SAR (Structure activity relationship) and overall biological verification are listed in their reports.

#3. There are many abbreviations without definition (DLK, ORF, PC6, NfL, RIPA, PVDF, HRP, and SNA). It is very hard for me to understand the abbreviations. As the instruction for Authors says “Acronyms/Abbreviations/Initialisms should be defined the first time they appear in each of three sections: the abstract; the main text; the first figure or table.”

Response 3: Thank you for pointing this out. And very sorry for the carelessness.  Therefore, l have rewritten the whole abbreviations in the revised manuscript.

#4. Why did the author select 174 and 331P1 from 6 compounds (Figure 1)?

Response 4: Thank you for pointing this out. 6 compounds were tested in the axotomy-induced degeneration, and the results showed that both 174 and 331P1 displayed potent protection, while other 4 compounds displayed weak protection of axon degeneration (supplementary Figure 1).

#5. Figure 4 does not prove anything. The authors need to show data from several doses of paclitaxel and vincristine. Also, please show quantitative data. It is very hard to compare expressions in Figure 4.

Response 5: Thank you for pointing this out. The rationale is that from the result showed in supplementary Figure 2 and Figure 3C that 2.0 μM PAC and 50 nM VCR induced severe damage to axons in the absence of SARM1 inhibitor compounds. The aim of this experiment was to verify whether the expression of SARM1 was changed when administrated with PAC or VCR.

supplementary Figure 2

Figure 3C

And it is good idea to show quantitative data about the results, in which I have inserted the the results in Figure 4C and 4D.

#6. I do not understand Table 2. I do not see the methods how the authors analyzed data (Methods section).

Response 6: Thank you for pointing this out. And very sorry for carelessness. Therefore, l have provided the methods needed in the revised manuscript.

#7. Discussion is also poor. There is no summary and limitations. The proposed presentation does not respect the elementary rules of a scientific writing. Additionally, the manuscript would benefit tremendously from language editing by either a native English speaker or a professional editor.

Response 7: Thank you for pointing this out. Therefore, l have rewritten the whole discussion in the revised manuscript. And we have sent our revised manuscript to a professional editor. And also, the new version of GA was presented as needed.

Minor comments

#8. The title is misleading. The authors evaluated some compounds not “a” compound.

Response 8: Thank you for pointing this out. l have edited the title in the revised manuscript.

#9. There are two definitions of degeneration index (Figure 3). “Axon degeneration index was calculated as the ratio of the degeneration axon over total axon area” (Page 3, Line 100) “Axon degeneration was quantified using a degeneration index, which ranges from 0 (perfectly intact) to 1 (perfectly fragmented)” (Page 10, Line 230). Which is correct?

Response 9: Thank you for pointing this out. The Axon degeneration index in Page 3, Line 100 refers to the result the degeneration was axotomy-induced. While Page 10 Line 259 (Line 230 in the last version of manuscript) refers to degeneration induced by vincristine treatment.

#10. What is compound “109A” (Page 6, Line 182)?

 Response 10:Sorry for the misunderstanding, it should be compound 160 for the mistake. In the .revised manuscript, it was revised as 160, as well as the Figure 2.

#11. Please show data low concentration to high concentrations (left to right) (Figure 3B).

 Response 11:Thank you for pointing this out. l have edited it in the revised manuscript.

#12. The authors need to show time course of NfL in Figure 5. Why did the authors select 15h?

 Response 12:Thank you for pointing this out. The previous reports showed that 15 h or 16h is mostly used in this assay. In this experiment, 15h was adopted in this assay.

#13. There is no method of table1.

Response 11:Thank you for pointing this out. l have written it in the revised manuscript. The assay is same with in vitro fluorescence assays of SARM1.

Comments on the Quality of English Language

The manuscript would benefit tremendously from language editing by either a native English speaker or a professional editor.

Submission Date

07 August 2024

Reviewer 2 Report

Comments and Suggestions for Authors

The manuscript by Chen and Li examined the protective effect of steril α and Toll/IL-1 receptor-motive containing 1 (SARM1) inhibitors in chemotherapy-induced peripheral neuropathy. The study is of interest and the compounds analysed demonstrate efficacy. However, the manuscript is written in a disorganised manner. It commences with the abstract, wherein it is unclear whether one or two compounds were analysed. The experiments conducted are logical, yet it is challenging to ascertain the rationale behind the specific experiments that were performed. The authors employ abbreviations without a clear definition of their meaning, which impedes comprehension of manuscript.

It is recommended that a detailed explanation of the experimental design be provided, including the rationale behind the specific experiments conducted, the criteria used to select the leading molecules, and the objective of the subsequent analysis. Moreover, a more comprehensive description of the methods would be beneficial.

Comments on the Quality of English Language

There is room for improvement in the English language.

Author Response

It is recommended that a detailed explanation of the experimental design be provided, including the rationale behind the specific experiments conducted, the criteria used to select the leading molecules, and the objective of the subsequent analysis. Moreover, a more comprehensive description of the methods would be beneficial.

Response : Thank you for you valuable advice. In the revised manuscript, we have edited the discussion in the revised manuscript. And we have sent our revised manuscript to a professional editor.

Round 2

Reviewer 1 Report

Comments and Suggestions for Authors

Comments and Suggestions for Authors

The authors moderately improved the manuscript. I still have, however, major concerns.

Major concerns

#1. Again, I’m not sure whether the protective effects of the compounds are through SARM1 (Figure 3, 5, and 6). The authors need to show that the effects are due to SARM1 inhibition.

#2. There is no summary and limitations in the Discussion.

Author Response

#1. Again, I’m not sure whether the protective effects of the compounds are through SARM1 (Figure 3, 5, and 6). The authors need to show that the effects are due to SARM1 inhibition.

Response 1: Thank you for your valuable advice and sorry for that I have not provided necessary explanation about this critical question. Indeed, we did not prove that the protection effect of our compounds are due to SARM1inhibition directly. From the Fig2, the results showed that TIR dimerization was disrupted by the compounds including 3311P1, thus inhibiting the activation of SARM1.

Also from the hydrolase inhibition effects in Fig1, we conclude that our compounds have inhibition towards SARM1.

Base on those results, the experiment of Fig3, 5 and 6 was done to explore the in vivo of 331P1 on protection of degenerated axons and CIPN. The results showed that 331P1 diplayed potent protective effect.

Besides, the structure of all the six compounds in this research are originated from the compounds in reference 21-22 (in the belowing fig). The detailed SAR (Structure activity relationship) and overall biological verification are listed in their reports (reference 21,22).

(cited from reference 22, https://doi.org/10.1016/j.neuron.2022.08.017 )

Based on those evidences, we may concluded that the protective effects of  compound 331P1 are due to SARM1 inhibition.

In the following research, we will continue to discuss the inhibitory role of 331P1 on SARM1 hydrolase activity in vivo model, in the hope of providing more direct evidence about that the inhibitary effects of 331P1 are due to SARM1 inhibition.

And once again, thanks for your instructive advice.

#2. There is no summary and limitations in the Discussion.

Response 2:Thank you for pointing this out and I have refreshed the revised part of summary and limitations in the latest version of manuscript.

  1. Summary and Limitations

A series of compounds for SARM1 inhibitors were designed based on the reported compounds, and compound 331P1 was identified with an IC50 of 189.3 nM against human SARM1 hydrolase and possessing in vitro pharmacokinetic properties. These features made compound 331P1 an ideal candidate for further in vivo studies. With compound 331P1, the protection of degenerated axons was verified in DRG model. The efficacy of compound 331P1 was also tested in SNA and CIPN of mouse model, the results showed that compound 331P1 restored plasma NFL levels, as expected for a SARM1 inhibitor. The mice with PAC-induced CIPN treated with compound 331P1 showed recovery of intraepidermal nerve fibers. These findings indicate that compound 331P1 is a selective and orally bioavailable SARM1 inhibitor with the potential for treating CIPN and other metabolic diseases associated with activated hydrolase activity of SARM1.

In conclusion, the most compelling evidence for the clinical translatability of compound 331P1 the demonstration of safe and efficacious neuroprotection in in vivo models of nerve injury. In this study, the efficacy of compound 331P1 was prioritized for further evaluation of their in vivo protection.

However, more research is needed to better understand the detailed molecular mechanism of downstream effectors, as well as more direct evidences about that the inhibitary effects of 331P1 are due to SARM1 inhibition. When it comes to 331P1 displaying potent protective effects against CIPN, it was found that calpain seems to be required for SARM1-dependent neuropathy in CIPN induced by both VCR and PAC (22,23,28,34). Deciphering whether calpain contributes to the protective effect of 331P1 in CIPN awaits further study.

Reviewer 2 Report

Comments and Suggestions for Authors

The authors have addressed all my questions. The manuscript can be published in itspresent form.

Author Response

The authors have addressed all my questions. The manuscript can be published in itspresent form.

Response : Thank you again for your time and having given me instructive advice about how to revise the manuscript.

Round 3

Reviewer 1 Report

Comments and Suggestions for Authors

I’m not convinced with the results that the protective effects of the compounds are through SARM1. I leave the decision to the editor. I have no further comments.